# Laboratory Tests, Bacterial Resistance, and Treatment Options in Adult Patients Hospitalized with a Suspected Urinary Tract Infection

**DOI:** 10.3390/diagnostics14111078

**Published:** 2024-05-22

**Authors:** Paul Froom, Zvi Shimoni

**Affiliations:** 1Clinical Utility Department, Sanz Medical Center, Laniado Hospital, Netanya 4244916, Israel; 2School of Public Health, University of Tel Aviv, Tel Aviv 6997801, Israel; 3The Adelson School of Medicine, Ariel University, Ariel 4070000, Israel; zshimoni@laniado.org.il; 4Sanz Medical Center, Laniado Hospital, Netanya 4244916, Israel

**Keywords:** urinary tract infection, internal medicine, laboratory tests, bacterial resistance, treatment options

## Abstract

Patients treated for systemic urinary tract infections commonly have nonspecific presentations, and the specificity of the results of the urinalysis and urine cultures is low. In the following narrative review, we will describe the widespread misuse of urine testing, and consider how to limit testing, the disutility of urine cultures, and the use of antibiotics in hospitalized adult patients. Automated dipstick testing is more precise and sensitive than the microscopic urinalysis which will result in false negative test results if ordered to confirm a positive dipstick test result. There is evidence that canceling urine cultures if the dipstick is negative (negative leukocyte esterase, and nitrite) is safe and helps prevent the overuse of urine cultures. Because of the side effects of introducing a urine catheter, for patients who cannot provide a urine sample, empiric antibiotic treatment should be considered as an alternative to culturing the urine if a trial of withholding antibiotic therapy is not an option. Treatment options that will decrease both narrower and wider spectrum antibiotic use include a period of watching and waiting before antibiotic therapy and empiric treatment with antibiotics that have resistance rates > 10%. Further studies are warranted to show the option that maximizes patient comfort and safety.

## 1. Introduction

Although urine tests are essential to diagnose a systemic urinary tract infection (UTI) [1], indiscriminate testing and the misunderstanding of the test results can lead to over and under-treatment, unnecessary introduction of a urinary catheter, more prolonged hospital stays, laboratory worker overload, and increased costs [2,3,4,5].

The proper testing and treatment of hospitalized adult patients with a suspected urinary tract infection requires considering that the presenting symptoms are nonspecific [6,7,8,9,10,11,12,13,14,15,16,17,18,19,20,21] and that positive urine tests do not confirm the diagnosis. There is no standardized combination of symptoms, signs, and laboratory test results to diagnose a urinary tract infection, and the diagnosis is uncertain unless both the blood and urine samples grow the same bacteria [19,20,21]. 

In the following narrative review, we will describe the widespread misuse of urine testing and consider how to limit testing and the disutility of urine cultures. Then, we will present the advantages and disadvantages of therapeutic options.

## 2. Tests Can Only Rule out a Urinary Tract Infection but Not Confirm the Diagnosis

There is a high prevalence of pyuria by dipstick and microscopy independent of the clinical presentation [6,18,22]. A study of elderly hospitalized patients reported a 42.4% prevalence of either a trace leukocyte esterase or positive nitrite on dipstick testing in those with a negative urine culture [23], and over 20 years ago, a study of incontinent but otherwise asymptomatic un-catheterized elderly institutionalized females reported pyuria defined as >10 WBC/HPF on microscopic examination in 59% of the specimens with bacteriuria and 34% of the specimens without bacteriuria [22]. Therefore, an abnormal urinalysis cannot confirm the diagnosis of a urinary tract infection.

Bacteriuria also cannot establish the diagnosis of a UTI since reported rates of asymptomatic bacteriuria are 16–18% in women older than age 70 years, 15–35% in institutionalized men, and 25–50% in institutionalized women [16,22,24,25,26,27]. In hospitalized febrile elderly patients with an extra-urinary tract reason for hospitalization, 30% of patients had bacteriuria [8].

## 3. The Role of the Dipstick to Rule out a Urinary Tract Infection

There might be a role for the urinalysis to rule out a UTI, reducing urine cultures and inappropriate hospitalizations and antibiotic therapy [23,28,29,30,31]. A meta-analysis reported in 2004 estimated that the sensitivity of the dipstick in detecting a UTI in inpatients was only 79% [32]. The studies included variable definitions of UTIs, variable performances of the dipstick, and differences in defining the cutoff for a positive urine culture. However, a positive dipstick (trace leukocyte esterase or positive nitrite test) identified 96.9% of elderly patients with a bacteremic UTI [23] and 90.1% of those with isolated bacteriuria [30]. Canceling orders for urine cultures would have decreased inappropriate antibiotic therapy and safely reduced urine cultures by 40% [23,30]. A study of non-surgical patients in the Netherlands found that reflex urine testing cancellation could prevent more than half of microscopic analysis and almost a fourth of urine cultures [31]. Because of the variability in the methods of dipstick testing, reflex cancelation of urine cultures requires ensuring that it is safe. 

## 4. A Positive Dipstick Should Not Be Confirmed by a Microscopic Urinalysis

The dipstick is a precise test [23,33,34,35] contrasted with the microscopic urinalysis that is not precise or accurate because of the effect of centrifugation, the variable urine volumes under the cover slip [36], and intra- and inter-observer variation [37,38] (Table 1). Centrifugation results in an average leukocyte loss of around 50% compared to counts in a Fuchs–Rosenthal counting chamber without centrifugation [36] and the volume under the cover slip can vary from 3- to 10-fold, dependent on the method of sampling and the physical properties of the urine. Furthermore, there can be differences in volumes of the centrifuged specimen and discarded supernatant, and in the thoroughness of mixing the specimen before and after centrifugation. Shimoni et al. [23] reported that confirmation of the dipstick with a microscopic examination would miss 21.9% of those with a bacteremic UTI.

The imprecision and inaccuracy of the microscopic urinalysis are often not appreciated and lead to diagnostic errors and selection bias [39,40]. A recent study in the United States gave physicians an option to proceed to a urine culture only if there were ten or more WBC/HPF; cultures decreased by nearly 50% adjusted for hospital days [39], but the disutility of canceling urine cultures inappropriately was not studied. Another study of patients with fever and a positive urine culture inappropriately included patients only if they had five WBC/HPF [40]. One way to limit microscopic urinalysis and diagnostic errors is to stop confirming the results of the dipstick by microscopy. Medical staff accepted the cessation of reflex microscopic examinations in outpatients [41,42] and inpatients [42]. Microscopic examinations done only by physician request decreased from 19,006 to less than 50 in the first study over a 6-month period [41] and from a mean of 3278 to 236 monthly tests in the other study [42].

## 5. Positive Findings on the Dipstick Can Lead to Inappropriate Further Testing

There is clinical utility if the dipstick rules out a urinary tract infection, but disutility includes reflex microscopic examinations and nephrological and urological referrals. Physician education is essential to limit orders of urinalysis, not reflex positive findings for microscopic examination, and to ignore the incidental finding of proteinuria and hematuria in hospitalized patients.

### 5.1. The Disutility of Finding Blood by Dipstick Analysis

Expert opinion still contends that a positive dipstick test for blood requires confirmation with a microscopic examination of the urine because of the possibility of a false-positive result due to the presence of semen, hemoglobinuria, myoglobinuria, or a high pH [43], but the reasons for the low precision of red blood cells/HPF is like that for WBC/HPF except that in addition preparation also causes various degrees of hemolysis [44]. Reagent strips for RBCs are more precise than microscopic urinalysis [33,45]. Therefore, microscopic urinalysis is not suitable to confirm dipstick hematuria.

Hematuria is common in hospitalized patients. In 1 study after the exclusion of patients with a urinary catheter or positive urine culture, 7.1% of all hospitalized patients and 24.4% of all patients undergoing microscopic urinalysis met the criteria for a urological workup [46]. However, a urological referral is of uncertain clinical utility and the incidental finding of blood on the dipstick in hospitalized patients should be ignored without microscopic confirmation [47].

### 5.2. Proteinuria

Dipstick proteinuria may be a sign of a renal disorder or associated with acute disease, and consequently, transient in hospitalized patients. Shimoni et al. [48] found that 45.0% of internal medicine patients with a urinalysis had proteinuria of ≥+1 (≥0.30 g/L). The results of the in-hospital urinalysis are unreliable because a negative dipstick test does not rule out proteinuria, and a positive test is likely to be transient or falsely positive. In patients with diabetes mellitus, the most common cause of end-stage kidney disease [49], it is important to test for proteinuria periodically in the outpatient setting because treatment can improve health outcomes. In the hospital setting dipstick proteinuria can lead to needless referrals and testing and should be ignored.

## 6. Other Tests to Rule out a Urinary Tract Infection

Other urine testing entails costs and increased laboratory technician time that is not always available, and there is no evidence that such testing improves the clinical utility of the automated dipstick test. Methods used to detect bacteriuria include gram stains, flow cytometry, digital imaging with particle recognition, automated microscopy with digital imaging [29,50,51,52,53,54], and manual determinations in a counting chamber such as the reference method of the Fuchs–Rosenthal chamber. The counting chamber is labor intensive, requires laboratory personnel with expertise and exposure to biological fluids, and is not in widespread use. Like the dipstick and microscopic urinalysis, other methodologies cannot be used to diagnose a urinary tract infection because of the highly prevalent abnormalities. The use of methods to rule out a urinary tract infection requires a high sensitivity to detect bacteriuria variously defined (see below) and that any false negatives do not result in patient morbidity or mortality.

Flow cytometers can detect bacteriuria and might perform better than the dipstick in ruling out the need for a urine culture. de Boer JF et al. [50] studied patients who presented to the emergency department with fever or with inflammatory markers and reported a 100% sensitivity with a specificity of 64.7% using the bacterial concentration measured on the Accuri C6 flow cytometer (BD Biosciences, San Jose, CA, USA) to identify urine cultures with at least 10^5^ colony forming units (CFU)/mL. There were similar findings in an earlier study of outpatients and inpatients [51]. Broeren et al. used the Sysmex UF-1000i (Medical Electronics, Kobe, Japan) [52] and reported a sensitivity of 96% and a specificity of 78% in identifying urine cultures with 10^5^ CFU/mL; the cut-off value was 230 bacteria/μL. Manoni et al. [53] chose cut-off values for bacteria and leukocytes of 125/μL and 40/μL, respectively, to obtain a sensitivity of 97% with a specificity of 94% with the Sysmex UF-1000i. Flow cytometers are more labor intensive but perform better than sediment analyzers [55]. It appears that the flow cytometers perform better than the dipstick in ruling out a UTI but require the demonstration that false negative test results do not endanger the patient before adaptation. Also, there is a need to balance patient benefits with increased costs and technician time compared to using automated dipstick analyzers.

## 7. Urine Cultures

There is a lack of consensus for urine culture microbial threshold recommendations for the clinical diagnosis of UTI; evidence is sparse and has changed over time. The most common definition of bacteriuria is ≥10^5^ CFU/mL [6,18,23], but others have used lower cut-offs: ≥10^4^ and even ≥10^2^ under certain circumstances. In 2012, the Society for Healthcare Epidemiology of America (SHEA) defined bacteriuria in cultures with only ≥10^2^ CFU/mL [14] in a sample obtained by a urinary catheterization procedure. In 2019, The Infectious Disease Society of America recommended using 10^5^ CFU/mL in voided urine as the threshold for defining true UTI in symptomatic patients [4], and the Centers for Disease Control and Prevention increased the minimum bacterial colony count from ≥10^4^ to ≥10^5^ CFU/mL. However, the UTI Reference Standard Consensus Group recently lowered the bacteriuria threshold from (10^5^ colony-forming units per mL) to 10^4^ colony-forming units per mL [56]. These are consensus decisions that are not evidence-based because the proper cut-off for diagnosing significant bacteriuria is uncertain. Higher cut-offs will decrease antibiotic use in those without a UTI but perhaps result in inappropriately withholding treatment.

### 7.1. Urine Cultures Are Overutilized and Have Clinical Disutility (Table 2)

Overuse of urine cultures in the hospital increases inappropriate use of antibiotics [8,57,58,59,60]. In patients with a positive urine culture, emergency department physicians in varied training stages often prescribe antibiotics to patients where clear, specific, and data-driven guidelines suggest treatment is unnecessary and potentially harmful [60]. A retrospective cohort study in 46 hospitals in the United States showed that 83% of patients with asymptomatic bacteriuria were treated with antibiotics [59] and had prolonged hospital stays. A 27% prevalence of urine cultures was reported in a national study of hospitalized patients of all ages in the United States, frequently in patients with reasons outside the urinary tract for admission (heart failure, acute myocardial infarction, cellulitis, and pneumonia) [61]. A meta-analysis of the literature using appropriateness of antimicrobial administration based on guidelines published by the Infectious Diseases Society of America found that isolation of gram-negative pathogens on urine culture, pyuria, nitrite positivity, and female sex increased the odds of receiving inappropriate treatment that occurred in 45% of patients with asymptomatic bacteriuria [62]. The authors emphasized, however, that the uncertainty in the differentiation between asymptomatic bacteriuria and a UTI partially explained the ‘overtreatment’ of patients with asymptomatic bacteriuria [62].

The use of antibiotics is the second most common cause of adverse drug event-related Emergency Department visits [63]; in hospitalized patients, 20% of the patients suffer at least one antibiotic-related adverse drug event within 90 days of antibiotic use, 97% of which results in added hospital days, clinic visits, or laboratory testing [64]. Patients prescribed antibiotics have an increased risk of a Clostridium difficile infection (CDI) or an infection due to antibiotic-resistant bacteria [65]; each day of antibiotic therapy increases individual risk of CDI by 9% [66]. Besides limiting their use, another intervention is shorter-course therapies. It was found in 1 study that 5–7 days of antibiotics is equivalent to longer durations [67] for patients with pyelonephritis. A step-down to a highly bioavailable oral agent [68] was successful in patients with bacteremia from a urinary source [68] as early as Day 3. Commonly cited key considerations for intravenous to oral conversion include clinical stability, absence of fever, and resolving leukocytosis, but it is unclear how to predict when step-down is safe. The reflex cancellation of the urine culture in those with a negative dipstick is one measure that will decrease the widespread inappropriate use of antibiotics.

The disutility of the urine culture also includes the need for urinary catheterization procedures to obtain a urine sample. In 1 study, 50% of the febrile elderly patients without another source for the infection needed urinary catheterization to obtain a sample for culturing [7], leading to an increased risk for an indwelling urinary catheter in the hospital and on discharge [7,8]. Thus, the balance between risks and benefits is unclear despite the consensus opinion recommending urinary catheterization in elderly febrile patients who cannot provide a urine specimen [69]. Finally, another potential harm of urine cultures is a false positive diagnosis that will lead to inappropriate antibiotic treatment and a delay in definitive interventions.

**Table 2 diagnostics-14-01078-t002:** Disutility of urine cultures.

1. Unnecessary use of antibiotics
a. Adverse drug events
b. Clostridium difficile infection
c. Increased bacterial resistance rates
2. Delay in definitive interventions
3. Introduction of a urine catheter to obtain a specimen

### 7.2. Bacterial Sensitivities and Antibiotic Therapy

We prefer that the initial treatment discordance between the antibiotic and bacterial sensitivity be called bacterial resistance to initial antibiotic therapy (BRIAT) and not inappropriate antibiotic therapy because the initial treatment might have been appropriate despite BRIAT.

There are claims that the accepted treatment for pyelonephritis resistance rates should be <10% [70,71,72]; the evidence for those recommendations is weak based on clinical experience, descriptive studies, and expert committee reports. It is unclear whether to accept a significant risk for BRIAT or to treat the patient with wider spectrum antibiotics that are generally held in reserve to decrease future resistance rates [73,74,75]. Risk factors for resistance, primarily from extended-spectrum beta-lactamase (ESBL) positivity include being bedridden, having a permanent urinary catheter as well as having a history of recent antibiotic usage, and having a previous hospitalization in an ESBL high-burden region [7,76,77]. One study reported that risk factors for BRIAT due to ESBL in elderly and stable internal medicine patients with a suspected UTI were male sex, previous hospitalization for a UTI, referral from a nursing home, and presence of a urinary catheter [78]. The risk of BRIAT increased from 13.0% to 69% as the number of risk factors increased from 0 to 4. The absence of risk factors, however, cannot rule out ESBL positivity and reported models to predict antibiotic susceptibility for urinary tract infections to third-generation cephalosporins such as ceftriaxone are poor with c-statistics <0.70 [7,79]. Mark et al. [40] reported ESBL infection risk factors in 82–81% of patients admitted with a febrile UTI in those with and without BRIAT. It appears that decisions on treatment cannot be made based on risk factors for resistant organisms.

Third-generation cephalosporins would not be used in most areas of the world as first-line empiric therapy for suspected urinary and respiratory tract infections if there was a requirement for <10% resistance rates. Ceftriaxone *Escherichia coli* resistance rates worldwide range from 5% to over 90% with a median of 45% [78,80,81]. Reports of ceftriaxone and other third-generation cephalosporins resistance rates vary from 12.9% in Northern California patients admitted with fever and a positive urine culture [40], 29% [79] of patients admitted to the hospital in Singapore and up to 43% of Turkish patients with pyelonephritis [82]. In Mexico [83] the prevalence of *Escherichia coli*-resistant organisms was 32.1%, and the average resistance rate for all the Enterobacteriaceae in University hospitals affiliated with the Center for Disease Control research network in the USA was 21% [84] ranging from around 5% to 45%. In Israel, resistance rates to Ceftriaxone were around 30% for *Escherichia coli* and 40–50% for *Klebsiella pneumonia* and *Proteus mirabilis* [80]. However, despite high resistance rates, many hospitals use empiric therapy with cephalosporins [40,80,81,82,85,86].

## 8. Should All Hospitalized Patients with a Suspected Systemic UTI Be Treated with Drugs That Have <10% Resistance Rates?

Although a chosen antibiotic may have been inadequate based on in vitro testing, it may have achieved sufficient therapeutic levels within the urinary tract to provide clinical benefit [87]; there may be a need to rethink urinary antibiotic breakpoints when treating multidrug-resistant organisms [88]. Urine concentrations, sometimes 100 to 1000 times higher than those achieved in the serum with equivalent doses can occur. For example, the peak ceftriaxone concentration after 1000 mg given intravenously is 151 mg/L in the serum but 995 mg/L in the urine. This may partially explain why two-thirds of elderly patients with BRIAT have reported responding [7,89,90] to initial therapy.

In patients with a UTI and systemic involvement but not presenting with septic shock, most studies have found that BRIAT does not significantly increase the mortality rates in patients with a febrile UTI [7], acute pyelonephritis [82,91], a complicated UTI [92], with urosepsis [78,84,93,94] or even in patients with a catheter-associated UTI [94]. A meta-analysis estimated that there was no definite increase in the risk for mortality in patients of all ages treated with BRIAT on internal medicine wards for upper UTIs [95]. There were no deaths due to BRIAT in 94 adults of all ages from Korea treated with either ceftriaxone or ciprofloxacin [91,96,97] and in 43 febrile females, half of whom were elderly treated with gentamycin [90]. On the other hand, Mark DG et al. [40] in Northern California studied patients admitted with a febrile UTI; patients with BRIAT had an increased 90-day mortality rate adjusted for comorbidities and severity of illness. However, for patients who die after 72 h, to show an association with BRIAT, the chart review needs to confirm that the death was due to end-organ damage, caused by the delay in “appropriate” antibiotic treatment. We are unaware of reports showing that BRIAT increases mortality in patients not presenting with septic shock.

There are hospitals where carbapenems cover nearly all bacteria grown in the urine [40,80,89], and upfront treatment in patients with a suspected UTI lowers the resistance rates to <10% and can shorten symptomatic times and hospitalization periods in patients by reducing patients with BRIAT [7,40,93,96]. The downside is the antibiotic treatment of patients without a UTI and increased overall resistance rates. To prepare for increasing rates of bacterial resistance to carbapenems, a recent randomized controlled trial in patients with systemic urinary tract infections found that an investigational drug Cefepime-Taniborbactam was non-inferior to meropenem with a 90% clinical response [98].

## 9. Watch and Wait

The best treatment option for symptomatic elderly clinically stable hospitalized patients without an extra-urinary tract presentation is uncertain. The knowledge that BRIAT does not reduce in-hospital survival or treatment success among stable, non-ICU patients may allow holding back antibiotic therapy in these patients until culture results are available [92]. This might reduce significantly antibiotic usage, since over >80% of elderly patients presenting to the emergency department with fever are hospitalized [99,100,101], and most are treated empirically with antibiotics. Around 40% were reported to have negative urine culture in 1 study [8]. An alternative approach to elderly hospitalized patients presenting with nonspecific symptoms without an extra-urinary tract source is to withhold antibiotic therapy if there is no systemic involvement [102] and no clinical predictors for ‘severe sepsis’ [103,104,105]. It is unclear, however, how to select patients for supportive therapy and monitoring without empirical antibiotics that are safe and specific enough to significantly decrease inappropriate antibiotic therapy. The decision requires balancing the side effects of antibiotics and hospitalization stays in patients without a bacterial infection and increased symptomatic periods and possible mortality in those with a UTI (Figure 1). Newer drugs could lend support for treating hospitalized patients with a suspected systemic UTI with drugs that have <10% resistance rates.

## 10. Conclusions

It is important to understand the limitations of urinalysis and consider options for the diagnosis and treatment of patients with a suspected UTI. Patients treated for systemic urinary tract infections commonly have nonspecific presentations, and the specificity of the results of the urinalysis and urine cultures is low. There are ways to limit testing, the disutility of urine cultures, and the use of antibiotics in hospitalized adult patients (Table 3). Automated dipstick testing is more precise and sensitive than microscopic urinalysis, which will result in false negative test results if ordered to confirm a positive dipstick test result. There is evidence that canceling urine cultures if the dipstick is negative (negative leukocyte esterase and nitrite) is safe and can decrease the overuse of urine cultures and unnecessary antibiotic therapy. Because of the side effects of introducing a urine catheter, for patients who cannot provide a urine sample, empiric antibiotic treatment should be considered as an alternative to culturing the urine if a trial of withholding antibiotic therapy is not an option. Treatment options that will decrease both narrower and wider spectrum antibiotic use include a period of watching and waiting before antibiotic therapy and empiric treatment with antibiotics that have resistance rates > 10%. Further studies are warranted to show which option maximizes patient comfort and safety.

## Figures and Tables

**Figure 1 diagnostics-14-01078-f001:**
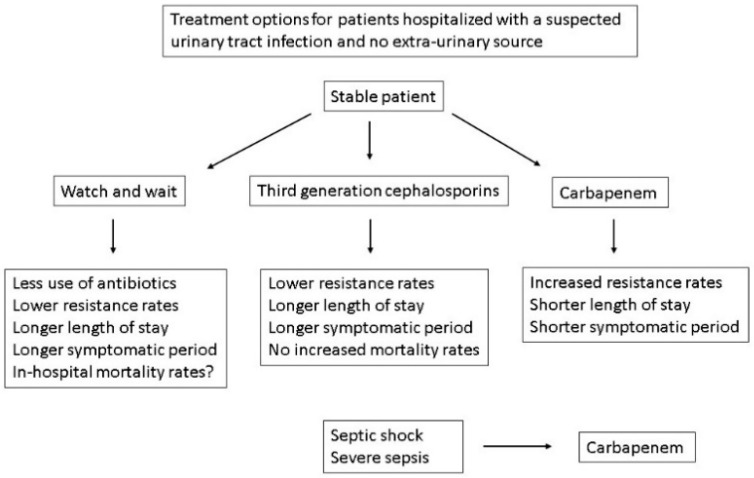
Benefits and risks of treatment options for hospitalized patients with a suspected urinary tract infection.

**Table 1 diagnostics-14-01078-t001:** Microscopic urinalysis causes of lack of precision and accuracy.

Variation in volume under the coverslip—3–10-fold
Variation in the discarded supernatant
Variation in mixing before and after centrifugation
Loss of cells during centrifugation
Intra and inter-observer variation

**Table 3 diagnostics-14-01078-t003:** Diagnosis and treatment of patients with a suspected urinary tract infection.

1. Limit urine cultures if the dipstick is negative.
2. Do not confirm the dipstick results with a microscopic urinalysis.
3. Limit reflexing incidental findings on dipstick (blood, protein).
4. Limit urine catheterization to obtain a urine sample.
5. Consider a watch and wait policy for stable patients with nonspecific symptoms.

## Data Availability

Not applicable.

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
