# Peer review of "Laboratory Tests, Bacterial Resistance, and Treatment Options in Adult Patients Hospitalized with a Suspected Urinary Tract Infection"

_diagnostics, 2024, doi:10.3390/diagnostics14111078_

Round 1

Reviewer 1 Report

Comments and Suggestions for Authors

The article highlights the diagnostic uncertainties surrounding urinary tract infections in hospitalized adult patients, emphasizing the limitations of laboratory tests like urine cultures and dipstick testing. It discusses treatment options, including watchful waiting, antibiotic therapy, and the challenges of using wide-spectrum antibiotics while considering patient safety and diagnostic uncertainties.

Find below some comments.

Line 38: Clarify the definition of "extended beyond the bladder" in the introduction to provide a clearer understanding of the diagnostic challenge with relevant references.

Line 51: Provide specific criteria for defining UTI symptoms/signs and laboratory test results to standardize diagnosis across studies and clinical settings with relevant references.

Line 67: provide recommendations on interpreting pyuria results in elderly hospitalized patients to improve diagnostic accuracy with relevant references.

Line 82: Discuss the potential role of dipstick testing in ruling out UTIs more effectively to reduce unnecessary hospitalizations and antibiotic prescriptions.

Line 96: Elaborate on the limitations of microscopic urinalysis compared to dipstick testing to highlight the importance of choosing the most accurate diagnostic method.

Line 107: Evaluate the utility of implementing criteria for proceeding to urine culture based on WBC count to optimize diagnostic efficiency while ensuring appropriate detection of UTIs.

Line 116: Clarify the implications of positive findings on urinalysis and provide guidance on avoiding unnecessary further testing, such as reflex microscopic examinations and referrals.

Line 122: Discuss the necessity of confirming positive dipstick tests for blood with microscopic examination, considering potential false-positive results and their implications for patient care. with relevant references

Line 130: Provide insights into the clinical significance of hematuria in hospitalized patients and guidance on appropriate management to avoid unnecessary referrals and interventions.

Line 136: Explore the significance of dipstick proteinuria in hospitalized patients, considering its potential causes and implications for patient care and outcomes.

Line 146: Evaluate the utility and limitations of other urine testing methods, such as gram stains and flow cytometry, in ruling out UTIs and discuss their impact on diagnostic accuracy and clinical decision-making.

Line 174: with relevant references Address the lack of consensus on urine culture microbial threshold recommendations and their implications for antibiotic use and patient care, emphasizing the need for evidence-based guidelines.

Line 231: Introduce the concept of bacterial resistance to initial antibiotic therapy (BRIAT) and distinguish it from inappropriate antibiotic therapy, emphasizing the importance of accurate terminology in clinical practice.

Line 236: Critically evaluate the evidence supporting the recommended resistance rates for pyelonephritis treatment and discuss the challenges in balancing the risk of bacterial resistance with the choice of antibiotic therapy.

Line 242: Discuss the risk factors associated with extended-spectrum beta-lactamase (ESBL) positivity in patients with suspected UTIs and the limitations of predicting antibiotic susceptibility based on these factors.

Line 254: Provide insights into global variations in ceftriaxone resistance rates among Escherichia coli and discuss their implications for empirical antibiotic therapy in different regions.

Line 268: with relevant references Evaluate the clinical significance of treating hospitalized patients with suspected complicated UTI/pyelonephritis with antibiotics having <10% resistance rates, considering the potential impact on patient outcomes and antibiotic resistance rates.

In the conclusion section: Addresses diagnostic uncertainty in UTI confirmation for elderly patients, advocating for prudent urine testing to prevent overuse and unnecessary treatments.

cross-check references used

A plagiarism check is recommended.

Comments on the Quality of English Language

Minor editing of English language required

Author Response

Please note that we have rewritten the manuscript and hope it is now acceptable.

Reviewer 2 Report

Comments and Suggestions for Authors

The authors of this manuscript raised issues in the laboratory diagnosis, bacteria resistance and treatment options in adult patients hospitalized with a suspected urinary tract infection.

The issues raised in this manuscript are clinically important. However, the manuscript needs to be written clearly and concisely.

Pyuria defined by either microscopy or dipstick is prevalent in the elderly and cannot confirm the diagnosis of a urinary tract infection. Automated dipstick testing is more precise and sensitive than the microscopic urinalysis which will result in false negative test results if ordered to confirm a positive dipstick test result.

The authors raised problems in the laboratory diagnosis of suspected urinary tract infections and treatment options in adult hospitalized patients.

All references are appropriate.

The manuscript is quite fragments, not well organized and structured. It's like putting pieces together. The logic and structure of the manuscripts need to be improved. The review needs to summarize the relevant evidence or findings or papers in a logical and restructured way.

Suggest having figures and tables to help readers understand.

The manuscript is hard to follow. In some very long sentences, meanings are unclear and hard to understand.

Comments on the Quality of English Language

The manuscripts need to be written clearly and concisely. In some very long sentences, meanings are unclear and hard to understand. 

Author Response

Please note that we have completely rewritten the article. 

Round 2

Reviewer 1 Report

Comments and Suggestions for Authors

no further comments 

Comments on the Quality of English Language

 English language fine. No issues detected

Author Response

Thank you for your review

Reviewer 2 Report

Comments and Suggestions for Authors

This manuscript is more like clinicians’ options than a review article.

The proper diagnosis of urine tract infections is problematic. The authors raised the issue of misusing laboratory tests for UTI diagnosis, including microscopic urinalysis and urine culture.

Negative microscopic urinalysis and urine culture results might be false negative due to sampling error or biofilm.

Don't understand why a positive urine culture not confirming the diagnosis. Do you mean although bacteria are present but do not cause infection?  The authors need to explain the reason and be supported by the evidence.

Need to explain the definition of clinically relevant urine tract infections.

The authors have strong opinions favour the urine dipstick test results. The authors need to explain the reason and be supported by the evidence. Urine Dipstick nitrite and leucocyte esterase positive results may be more clinically relevant.

Comments on the Quality of English Language

English language is still hard to understand and needs to be improved to be written clearly.

Author Response

  1. This manuscript is more like clinicians’ options than a review article.

  That's why we called it a narrative review.

2. The proper diagnosis of urine tract infections is problematic. The authors raised the issue of misusing laboratory tests for UTI diagnosis, including microscopic urinalysis and urine culture.

We agree

3. Negative microscopic urinalysis and urine culture results might be false negative due to sampling error or biofilm.

We agree with that.

4. Don't understand why a positive urine culture not confirming the diagnosis. Do you mean although bacteria are present but do not cause infection?  The authors need to explain the reason and be supported by the evidence.

Please refer to section 2. " Bacteriuria also cannot establish the diagnosis of a UTI since reported rates of asymptomatic bacteriuria are 16–18% in women older than age 70 years, 15–35% in institutionalized men, and 25–50% in institutionalized women [16,22,24-27]. In hospitalized febrile elderly patients with an extra-urinary tract reason for hospitalization, 30% of patients had bacteriuria [8].    

5. Need to explain the definition of clinically relevant urine tract infections.

Treating a symptomatic urinary tract infection is clinically relevant.

5. The authors have strong opinions favour the urine dipstick test results. The authors need to explain the reason and be supported by the evidence. Urine Dipstick nitrite and leucocyte esterase positive results may be more clinically relevant.

This is explained and the evidence is given. Please see sections two and three.

5. Comments on the Quality of English Language English language is still hard to understand and needs to be improved to be written clearly.

We respectfully disagree.